# PeerJ

# Snake venomics of *Bothrops punctatus*, a semi-arboreal pitviper species from Antioquia, Colombia

Maritza Fernández Culma[1], Jaime Andrés Pereañez[1,2], Vitelbina Núñez Rangel[1,3] and Bruno Lomonte[4]

[1] Programa de Ofidismo/Escorpionismo, Universidad de Antioquia UdeA, Medellín, Colombia
[2] Facultad de Química Farmacéutica, Universidad de Antioquia UdeA, Medellín, Colombia
[3] Escuela de Microbiología, Universidad de Antioquia UdeA, Medellín, Colombia
[4] Instituto Clodomiro Picado, Facultad de Microbiología, Universidad de Costa Rica, San José, Costa Rica

## ABSTRACT

*Bothrops punctatus* is an endangered, semi-arboreal pitviper species distributed in Panamá, Colombia, and Ecuador, whose venom is poorly characterized. In the present work, the protein composition of this venom was profiled using the 'snake venomics' analytical strategy. Decomplexation of the crude venom by RP-HPLC and SDS-PAGE, followed by tandem mass spectrometry of tryptic digests, showed that it consists of proteins assigned to at least nine snake toxin families. Metalloproteinases are predominant in this secretion (41.5% of the total proteins), followed by C-type lectin/lectin-like proteins (16.7%), bradykinin-potentiating peptides (10.7%), phospholipases $A_2$ (9.3%), serine proteinases (5.4%), disintegrins (3.8%), L-amino acid oxidases (3.1%), vascular endothelial growth factors (1.7%), and cysteine-rich secretory proteins (1.2%). Altogether, 6.6% of the proteins were not identified. *In vitro*, the venom exhibited proteolytic, phospholipase $A_2$, and L-amino acid oxidase activities, as well as angiotensin-converting enzyme (ACE)-inhibitory activity, in agreement with the obtained proteomic profile. Cytotoxic activity on murine C2C12 myoblasts was negative, suggesting that the majority of venom phospholipases $A_2$ likely belong to the acidic type, which often lack major toxic effects. The protein composition of *B. punctatus* venom shows a good correlation with toxic activities here and previously reported, and adds further data in support of the wide diversity of strategies that have evolved in snake venoms to subdue prey, as increasingly being revealed by proteomic analyses.

## INTRODUCTION

The Chocoan forest lancehead, *Bothrops punctatus*, known in Colombia as 'rabo de chucha', is a large semi-arboreal pitviper, ranging from 1.0 to 1.5 m in length. *Campbell & Lamar (2004)* described its distribution from the Pacific foothills and coastal plain of eastern Panamá through western Colombia to northwestern Ecuador, with an altitudinal range between 1350 and 2300 m. In Colombia, *Daza, Quintana & Otero (2005)* reported

Corresponding author
Bruno Lomonte,
bruno.lomonte@ucr.ac.cr

the occurence of *B. punctatus* in the Cauca and Magdalena river basins of Antioquia to eastern Chocó. Although *Bothrops* species are clearly predominant in the epidemiology of snakebite accidents occuring in Colombia (*Otero, 1994*; *Paredes, 2012*), published reports of proven envenomings caused by *B. punctatus* appear to be rare. The protein composition of the venom of this species has not been investigated, although at least two reports characterized its toxicological properties, in comparative studies of snake venoms from Colombia (*Otero et al., 1992*) and Ecuador (*Kuch et al., 1996*), respectively. The lethal potency of this venom to mice was highest among the different *Bothrops* venoms analyzed in these two studies, being only second to that of *Crotalus durissus terrificus* venom (*Otero et al., 1992*; *Kuch et al., 1996*). Due to the lack of knowledge on the venom composition of *B. punctata*, this work aimed at characterizing its proteomic profile using the 'snake venomics' analytical strategy (*Calvete, Juárez & Sanz, 2007*; *Calvete, 2011*), in combination with the assessment of its enzymatic or toxic activities *in vitro*.

## METHODS

### Venom

Venom was obtained from two adult *Bothrops punctatus* specimens collected in the eastern region of the Department of Antioquia, and kept in captivity at the Serpentarium of Universidad de Antioquia, Medellín, Colombia, under institutional permission for Programa de Ofidismo/Escorpionismo. Venom samples were centrifuged to remove debris, pooled, lyophilized and stored at −20°C. In some functional assays, pooled venom obtained from more than 30 specimens of *Bothrops asper*, collected in the Departments of Antioquia and Chocó, was included for comparative purposes.

### Proteomic profiling

For reverse-phase (RP) HPLC separations, 2.5 mg of venom was dissolved in 200 μL of water containing 0.1% trifluoroacetic acid (TFA; solution A), centrifuged for 5 min at $15,000 \times g$, and loaded on a $C_{18}$ column ($250 \times 4.6$ mm, 5 μm particle; Teknokroma) using an Agilent 1200 chromatograph with monitoring at 215 nm. Elution was performed at 1 mL/min by applying a gradient towards solution B (acetonitrile, containing 0.1% TFA), as follows: 0% B for 5 min, 0–15% B over 10 min, 15–45% B over 60 min, 45–70% B over 10 min, and 70% B over 9 min (*Lomonte et al., 2014*). Fractions were collected manually, dried in a vacuum centrifuge, and further separated by SDS-PAGE under reducing or non-reducing conditions, using 12% gels. Protein bands were excised from Coomassie blue R-250-stained gels and subjected to reduction with dithiothreitol (10 mM) and alkylation with iodoacetamide (50 mM), followed by in-gel digestion with sequencing grade bovine trypsin (in 25 mM ammonium bicarbonate, 10% acetonitrile) overnight on an automated processor (ProGest Digilab), according to the manufacturer. The resulting peptide mixtures were analyzed by MALDI-TOF-TOF mass spectrometry on an Applied Biosystems 4800-Plus instrument. Peptides were mixed with an equal volume of saturated $\alpha$-CHCA matrix (in 50% acetonitrile, 0.1% TFA), spotted (1 μL) onto Opti-TOF 384-well plates, dried, and analyzed in positive reflector mode. Spectra were acquired using a

**Peer**J ________________________________________________

laser intensity of 3000 and 1500 shots/spectrum, using as external standards CalMix-5 (ABSciex) spotted on the same plate. Up to 10 precursor peaks from each MS spectrum were selected for automated collision-induced dissociation MS/MS spectra acquisition at 2 kV, in positive mode (500 shots/spectrum, laser intensity of 3000). The resulting spectra were analyzed using ProteinPilot v.4 (ABSciex) against the UniProt/SwissProt database using the Paragon® algorithm at a confidence level of ≥95%, for the assignment of proteins to known families. Few peptide sequences with lower confidence scores were manually searched using BLAST (http://blast.ncbi.nlm.nih.gov). Finally, the relative abundance of each protein (% of total venom proteins) was estimated by integration of the peak signals at 215 nm, using Chem Station B.04.01 (Agilent). When a peak from HPLC contained two or more SDS-PAGE bands, their relative distribution was estimated by densitometry using the Image Lab v.2.0 software (Bio-Rad) (*Calvete, 2011*).

## Venom activities

### Phospholipase A$_2$ activity

Venom phospholipase A$_2$ (PLA$_2$) activity was determined on the monodisperse synthetic substrate 4-nitro-3-octanoyl-benzoic acid (NOBA) (*Holzer & Mackessy, 1996*), in triplicate wells of microplates. Twenty μL of venom solutions, containing 20 μg protein, were mixed with 20 μL of water, 200 μL of 10 mM Tris, 10 mM CaCl$_2$, 100 mM NaCl, pH 8.0 buffer, and 20 μL of NOBA (0.32 mM final concentration). Plates were incubated at 37°C, and the change in absorbance at 425 nm was recorded after 20 min in a microplate reader (Awareness Technology).

### Proteolytic activity

Proteolysis was determined upon azocasein (Sigma-Aldrich) as described by *Wang, Shih & Huang (2004)*. Twenty μg of venoms were diluted in 20 μL of 25 mM Tris, 0.15 M NaCl, 5 mM CaCl$_2$, pH 7.4 buffer, added to 100 μL of azocasein (10 mg/mL) and incubated for 90 min at 37°C. The reaction was stopped by adding 200 μL of 5% trichloroacetic acid. After centrifugation, 100 μL of supernatants were mixed with an equal volume of 0.5 M NaOH, and absorbances were recorded at 450 nm. Experiments were carried out in triplicate.

### L-amino acid oxidase activity

L-amino acid oxidase (LAAO) activity was determined by adding various concentrations of venom (2.5–20 μg) in 10 μL of water to 90 μL of a reaction mixture containing 250 mM ʟ-Leucine, 2 mM *o*-phenylenediamine, and 0.8 U/mL horseradish peroxidase, in 50 mM Tris, pH 8.0 buffer, in triplicate wells of a microplate (*Kishimoto & Takahashi, 2001*). After incubation at 37°C for 60 min, the reaction was stopped with 50 μL of 2 M H$_2$SO$_4$, and absorbances were recorded at 492 nm.

### Cytotoxic activity

Cytotoxic activity was assayed on murine skeletal muscle C2C12 myoblasts (ATCC CRL-1772) as described by *Lomonte et al. (1999)*. Venom (40 μg) was diluted in assay medium (Dulbecco's Modified Eagle's Medium [DMEM] supplemented with 1% fetal

calf serum [FCS]), and added to subconfluent cell monolayers in 96-well plates, in 150 µL, after removal of growth medium (DMEM with 10% FCS). Controls for 0 and 100% toxicity consisted of assay medium, and 0.1% Triton X-100 diluted in assay medium, respectively. After 3 h at 37°C, a supernatant aliquot was collected to determine the lactic dehydrogenase (LDH; EC 1.1.1.27) activity released from damaged cells, using a kinetic assay (Wiener LDH-P UV). Experiments were carried out in triplicate.

### ACE inhibitory activity

The angiotensin-converting enzyme (ACE) inhibitory activity of fraction 4 from the HPLC separation (see Table 1), which was identified as a bradykinin-potentiating peptide-like component, was assayed by the method of *Cushman & Cheung (1971)* with some modifications (*Kim et al., 1999*). Various concentrations of the fraction, diluted in 20 µL, were added to 100 µL of 10 mM N-hippuryl-His-Leu substrate diluted in 2 mM potassium phosphate, 0.6 M NaCl, pH 8.3 buffer, and 5 mU of ACE (EC 3.4.15.1; 5.1 UI/mg) diluted in 50% glycerol. The reaction was incubated at 37°C for 30 min, and stopped by adding 200 µL of 1 NHCl. The produced hippuric acid was extracted by vigorous stirring for 10 s, followed by the addition of 600 µL of ethyl acetate, and centrifugation for 10 min at $4000 \times g$. An aliquot of 500 µL of organic phase was dried at 95°C for 10 min. The residue was dissolved in 1 mL of water and, after stirring, the absorbance was measured at 228 nm. The percentage of ACE inhibition (% ACEi) was determined using the following formula; % ACEi = (Abs Control − Abs sample)/(Abs control − Abs blank). Control absorbance corresponded to hippuric acid formed after the action of ACE, while blank absorbance was enzyme without substrate.

### Statistical analyses

The significance of differences between means was assessed by ANOVA, followed by Dunnett's test, when several experimental groups were compared with the control group, or by Student's t-test, when two groups were compared. Differences were considered significant if $p < 0.05$.

## RESULTS AND DISCUSSION

*B. punctatus* has been included in the 'red list', a report categorizing conservation status, as a threatened species (*Carrillo et al., 2005*). Very scarce information on its venom is available in the literature. In comparative studies of snake venoms from Colombia (*Otero et al., 1992*) and Ecuador (*Kuch et al., 1996*), respectively, this venom was found to induce local effects such as hemorrhage, edema, and myonecrosis, as well as systemic alterations such as defibrination, in similarity to venoms from other *Bothrops* species. Developments in proteomic techniques have brought new possibilities to examine the detailed toxin composition of snake venoms, increasing knowledge on their evolution, toxicological properties, and correlation with clinical features of envenomings (*Calvete, Juárez & Sanz, 2007*; *Calvete, 2013*; *Fox & Serrano, 2008*; *Valente et al., 2009*; *Ohler et al., 2010*). Therefore, the venom of *B. punctatus* was analyzed for the first time using proteomic tools, to gain a deeper understanding on its protein composition and relationships to toxic and enzymatic actions.

**Table 1** Assignment of the RP-HPLC isolated fractions of *Bothrops punctatus* venom to protein families by MALDI-TOF-TOF of selected peptide ions from in-gel trypsin-digested protein bands.

| Peak | % | Mass (kDa) | Peptide ion | | MS/MS-derived amino acid sequence[*] | Protein family; ∼ related protein |
|---|---|---|---|---|---|---|
| | | | *m/z* | *z* | | |
| 1 | 0.2 | | - | - | - | unknown |
| 2 | 0.3 | | - | - | - | unknown |
| 3 | 1.6 | | - | - | - | unknown |
| 4 | 10.7 | - | 967.5 | 1 | ZBWAPVBK | BPP-like; ∼ Q7T1M3 |
| 5 | 0.8 | ▼10 | 2259.1 | 1 | XARGDDM$^{ox}$ DDYCNGXSAGCPR | Disintegrin; ∼ Q7SZD9 |
| | | | 2051.0 | 1 | XRPGABCAEGXCCDBCR | |
| | | | 2459.0 | 1 | EAGEECDCGTPGNPCCDAATCK | |
| 6 | 3.0 | ▼10 | 1902.9 | 1 | GDDMDDYCNGXSAGCPR | Disintegrin; ∼ Q0NZX5 |
| | | | 2243.1 | 1 | XARGDDMDDYCNGXSAGCPR | |
| | | | 2051.0 | 1 | XRPGABCAEGXCCDBCR | |
| | | | 2459.1 | 1 | EAGEECDCGTPGNPCCDAATCK | |
| 7 | 0.3 | | - | - | - | unknown |
| 8 | 1.7 | ▼11 | 2062.0 | 1 | CGGCCTDESXECTATGBR | VEGF; ∼ Q90X23 |
| | | | 3134.9 | 1 | ETXVSXXEEHPDEVSHXFRPSCVTAXR | |
| 9 | 1.2 | ▼22 ■18 | 2526.1 | 1 | SGPPCGDCPSACDNGXCTNPCTK | CRISP; ∼ Q7ZT99 |
| | | | 1537.8 | 1 | MEWYPEAAANAER | |
| | | | 1828.9 | 1 | YFYVCBYCPAGNMR | |
| 10a | 0.4 | ▼38 | 1561.9 | 1 | SVPNDDEEXRYPK | Serine proteinase; ∼ Q5W960 |
| 10b | 0.2 | ▼29 ■28 | 1206.8 | 1 | XMGWGTXSPTK | Serine proteinase; ∼ Q072L6 |
| | | | 1683.2 | 1 | TYTBWDBDXMXXR | |
| | | | 2534.5 | 1 | VSYPDVPHCANXNXXDYEVCR | |
| | | | 1069.8 | 1 | FXVAXYTSR | |
| | | | 1512.8 | 1 | VXGGDECNXNEHR | |
| | | | 3387.8 | 1 | DSCBGDSGGPXXCNGBFBGXXSWGVHPCGBR | |
| 10c | 0.3 | ▼12 ■22 | - | - | - | unknown |
| 11 | 1.5 | ▼28 ■20 | 1288.7 | 1 | NFBMBXGVHSK | Serine proteinase; ∼ Q072L6 |
| | | | 1190.7 | 1 | XMGWGTXSPTK | |
| | | | 2305.4 | 1 | AAYPWBPVSSTTXCAGXXBGGK | |
| | | | 1140.6 | 1 | VSDYTEWXK | |
| | | | 2477.5 | 1 | VSNSEHXAPXSXPSSPPSVGSVCR | |
| | | | 2477.4 | 1 | VXGGDECNXNEHR | |
| 12a | 1.8 | ▼35 | 1083.7 | 1 | FXAFXYPGR | Serine proteinase; ∼ Q6IWF1 |
| 12b | 0.4 | ▼29 ■22 | 1517.9 | 1 | NDDAXDBDXMXVR | Serine proteinase; ∼ Q5W959 |
| | | | 1499.8 | 1 | VVGGDECNXNEHR | |
| | | | 2294.3 | 1 | TNPDVPHCANXNXXDDAVCR | |
| | | | 1279.7 | 1 | AAYPEXPAEYR | |
| | | | 2889.7 | 1 | XDSPVSNSEHXAPXSXPSSPPSVGSVCR | |
| | | | 1083.7 | 1 | FXAFXYPGR | |
| 13–15 | 0.8 | | - | - | - | unknown |

Table 1 (*continued*)

| Peak | % | Mass (kDa) | Peptide ion | | MS/MS-derived amino acid sequence[*] | Protein family; ~ related protein |
|------|---|------------|-------------|---|--------------------------------------|-----------------------------------|
| | | | *m/z* | *z* | | |
| 16 | 3.1 | ▼16 ■16 | 1505.7 | 1 | CCFVHDCCYGK | Phospholipase $A_2$, D49; ~ P86389 |
| | | | 934.6 | 1 | YWFYGAK | |
| | | | 1966.1 | 1 | YXSYGCYCGWGGXGBPK | |
| | | | 2064.1 | 1 | DATDRCCFVHDCCYGK | |
| | | | 2027.2 | 1 | DNBDTYDXBYWFYGAK | |
| | | | 2626.4 | 1 | XDXYTYSBETGDXVCGGDDPCBK | |
| | | | 1786.0 | 1 | BXCECDRVAATCFR | |
| 17a | 0.4 | ▼14 ■21 | 1928.9 | 1 | DCPPDWSSYEGHCYR | C-type lectin/lectin-like; ~ P22030 |
| 17b | 1.7 | ▼15 ■16 | 2027.1 | 1 | DNBDTYDXBYWFYGAK | Phospholipase $A_2$, D49; ~ C9DPL5 |
| 17c | 0.4 | ■13 | 1720.8 | 1 | E$^{pa}$ NGDVVCGGDDPCBK | Phospholipase $A_2$, D49; ~ P86389 |
| | | | 1505.7 | 1 | CCFVHDCCYGK | |
| | | | 2064.0 | 1 | DATDRCCFVHDCCYGK | |
| 18 | 2.8 | ▼13 | 2064.0 | 1 | DATDRCCFVHDCCYGK | Phospholipase $A_2$, D49; ~ Q9I968 |
| 19 | 0.3 | | - | - | - | unknown |
| 20 | 6.2 | ▼13 ■19 | 1928.9 | 1 | DCPSDWSPYEGHCYR | C-type lectin/lectin-like; ~ Q9PS06 |
| 21 | 0.8 | | - | - | - | unknown |
| 22a | 0.9 | ■120 | 1537.8 | 1 | ACSNGBCVDVNRAS | Metalloproteinase; ~ Q8AWI5 |
| | | | 1269.7 | 1 | SAECTDRFBR | |
| 22b | 3.1 | ▼53 ■48 | 3185.9 | 1 | VVXVGAGMSGXSAAYVXANAGHBVTVXEASER | L-amino acid oxidase; ~ Q6TGQ9 |
| | | | 2605.5 | 1 | BFGXBXNEFSBENENAWYFXK | |
| | | | 2271.3 | 1 | XYFAGEYTABAHGWXDSTXK | |
| | | | 1388.8 | 1 | BFWEDDGXHGGK | |
| | | | 1352.8 | 1 | SAGBXYEESXBK | |
| 22c | 0.9 | ▼13 | 1636.0 | 1 | NXBSSDXYAWXGXR | C-type lectin/lectin-like; ~ P22029 |
| | | | 1928.9 | 1 | DCPPDWSSYEGHCYR | |
| 23–25a | 1.3 | ▼13 | 1533.7 | 1 | SYGAYGCNCGVXGR | Phospholipase $A_2$, K49; ~ Q9PVE3 |
| 23–25b | 1.1 | ▼28, ■20 | 1279.7 | 1 | AAYPEXPAEYR | Serine proteinase; ~ Q5W959 |
| | | | 14.997 | 1 | VVGGDECNXNEHR | |
| | | | 2294.1 | 1 | TNPDVPHCANXNXXDDAVCR | |
| 23–25c | 0.9 | ▼13, ■19 | 1635.8 | 1 | NXBSSDXYAWXGXR | C-type lectin/lectin-like; ~ P22029 |
| 26 | 14.4 | ▼23 ■42 | 2040.2 | 1 | YXYXDXXXTGVEXWSNK | Metalloproteinase; ~ P86976 |
| | | | 1114.6 | 1 | XHBMVNXMK | |
| | | | 2257.3 | 1 | DXXNVBPAAPBTXDSFGEWR | |
| | | | 1828.0 | 1 | YVEXFXVVDHGMFMK | |
| 27 | 2.0 | | - | - | - | unknown |
| 28a | 18.3 | ▼46 ■42 | 1552.7 | 1 | VCSNGHCVDVATAY | Metalloproteinase; ~ Q8QG88 |
| | | | 2953.3 | 1 | ASM$^{ox}$ SECDPAEHCTGBSSECPADVFHK | |
| | | | 2154.2 | 1 | XTVBPDVDYTXNSFAEWR | |

Table 1 (*continued*)

| Peak | % | Mass (kDa) | Peptide ion | | MS/MS-derived amino acid sequence* | Protein family; ∼ related protein |
|------|---|-----------|-------------|---|------------------------------------|-----------------------------------|
| | | | *m/z* | *z* | | |
| 28b | 2.1 | ■21 | 3261.7 | 1 | TDXVSPPVCGNYFVEVGEDCDCGSPATCR | Metalloproteinase; |
| | | | 1457.0 | 1 | XVXVADYXM$^{ox}$ FXK | ∼ O93517 |
| 28c | 6.2 | ▼14 | 1635.9 | 1 | NXBSSDXYAWXGXR | C-type lectin-like; |
| | | | 1193.6 | 1 | TTDNBWWSR | ∼ P22029 |
| 29a | 3.2 | ▼46 | 2154.2 | 1 | XTVBPDVDYTXNSFAEWR | Metalloproteinase; |
| | | | 1609.9 | 1 | XYEXVNTXNVXYR | ∼ Q8QG88 |
| | | | 1775.0 | 1 | YVEFFXVVDBGMVTK | |
| 29b | 2.1 | ▼14 | 992.5 | 1 | MNWADAER | C-type lectin/lectin-like; |
| | | | 1928.8 | 1 | DCPPDWSSYEGHCYR | ∼ M1V359 |
| | | | 1842.9 | 1 | MNWADAERFCSEQAK | |
| 30 | 2.6 | ▼38 | 1327.8 | 1 | YXEXVXVADHR | Metalloproteinase; ∼ Q8AWX7 |

**Notes.**

* Cysteine residues determined in MS/MS analyses are carbamidomethylated.

X, Leu/Ile; B, Lys/Gln; $^{ox}$, oxidized; $^{pa}$, propionamide.

▼, reduced, or ■, non-reduced SDS-PAGE mass estimations, in kDa. Abbreviations for protein families as in Fig. 2.

RP-HPLC of the crude venom resulted in the separation of 30 fractions (Fig. 1C), which were further subjected to SDS-PAGE (Fig. 1B), in-gel digestion of the excised bands, and MALDI-TOF-TOF analysis of the resulting peptides. The amino acid sequences obtained allowed the unambiguous assignment of 29 out of the 37 components analyzed, to known protein families of snake venoms (Table 1). Protein family relative abundances were estimated by integration of the chromatographic areas, combined with gel densitometric scanning. Results showed that the predominant proteins in this secretion are metalloproteinases (41.5%; SVMP), followed by C-type lectin/lectin-like proteins (16.7%; CTL), bradykinin-potentiating peptide-like peptides (10.7%; PEP), phospholipases $A_2$ of both the D49 (8.0%) and K49 (1.3%) subtypes (for a combined 9.3%; $PLA_2$), serine proteinases (5.4%; SP), disintegrins (3.8%; DIS), L-amino acid oxidases (3.1%; LAO), vascular endothelial growth factor (1.7%; VEGF), and cysteine-rich secretory proteins (1.2%; CRISP), as summarized in Fig. 2 and Table 1. An estimated 6.6% of the proteins remained unidentified, and owing to the scarcity of the venom, their assignment could not be further pursued.

A recent phylogenetic analysis of the genus *Bothrops* (*sensu lato*) by *Fenwick et al. (2009)* grouped *B. punctatus* within the same clade as *Bothrops atrox* and *Bothrops asper*. Since the proteomic profile of the venoms of the latter two species has been reported (*Núñez et al., 2009*; *Alape-Girón et al., 2008*), a comparison of their venom compositions, together with those of two other pitviper species distributed in Colombia, *Bothrops ayerbei* (*Mora-Obando et al., 2014*) and *Bothriechis schlegelii* (*Lomonte et al., 2008*), was compiled (Table 2). Venoms from these five species have been analyzed by the same methodological strategy, therefore allowing reliable comparisons. The composition of *B. punctatus* venom resembles that of the other *Bothrops* species listed in Table 2 only

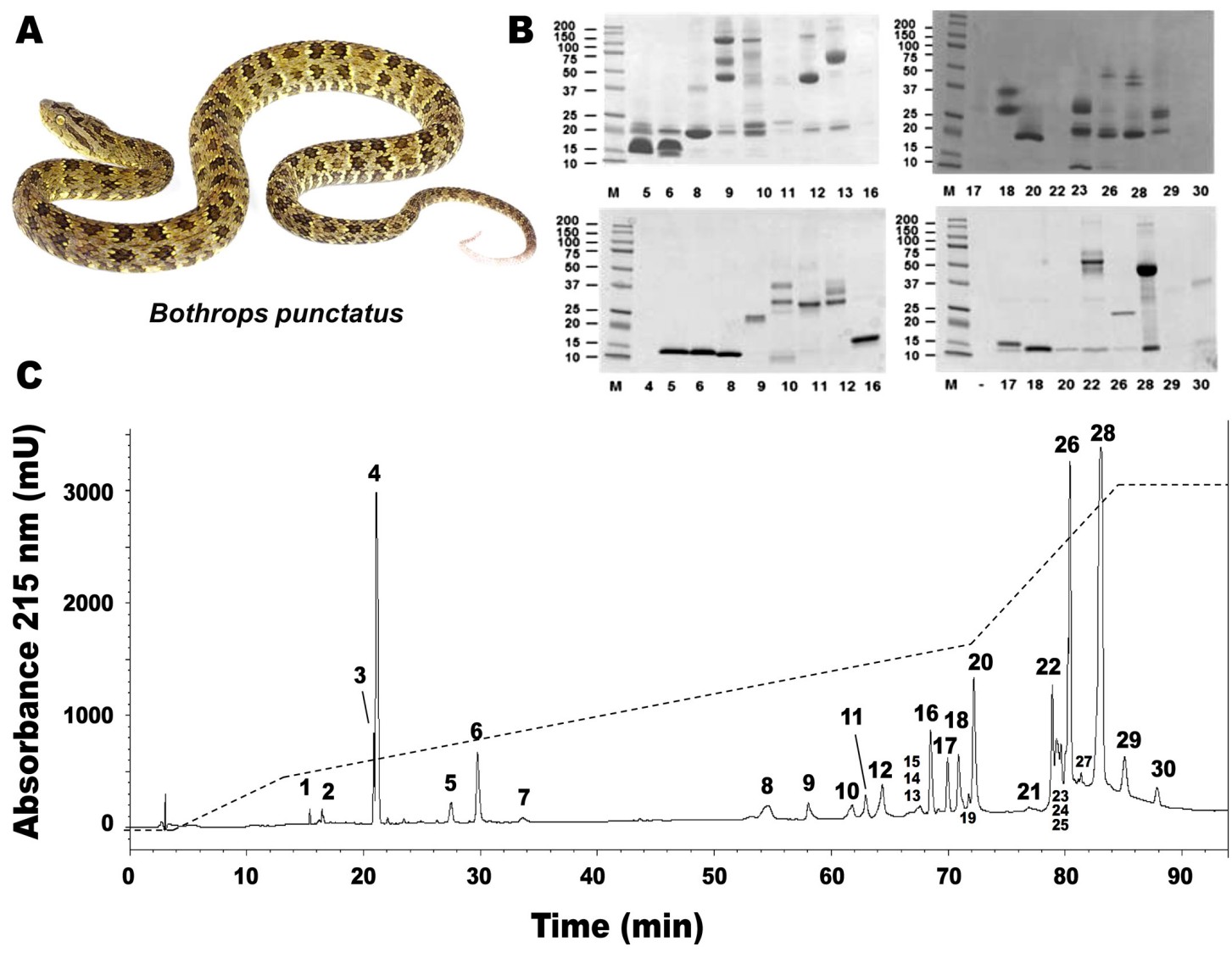

**Figure 1 Separation of *Bothrops punctatus* (A) venom proteins by RP-HPLC (C) and SDS-PAGE (B).** Venom was fractionated on a $C_{18}$ column (C) by applying an acetonitrile gradient from 0 to 70% (dashed line), as described in Methods. Each fraction was analyzed by SDS-PAGE (B) under non-reducing (top gels) or reducing (bottom gels) conditions. Molecular weight markers (M) are indicated in kDa, at the left. Tryptic digests of the excised protein bands were characterized by MALDI-TOF/TOF, as summarized in Table 1. The photograph of *B. punctatus* was obtained with permission from www.tropicalherping.com.

in terms of their high content of metalloproteinases (41.5–53.7%), but overall, its composition departs from the relative protein abundances observed in any of the other four pitvipers. The high proportion of CTL proteins in *B. punctatus* is of note, doubling the abundance observed in *B. atrox*, and close to that of *B. ayerbei*, while in contrast such proteins are expressed only in trace amounts in *B. asper*, and have not been detected in *B. schlegelii* (Table 2). Further, *B. punctatus* venom presents a modest amount of VEGF (1.7%), which has not been found in any of the venoms listed in Table 2. Similar to the venom of the arboreal snake *B. schlegelii*, but also with the terrestrial species *B.*

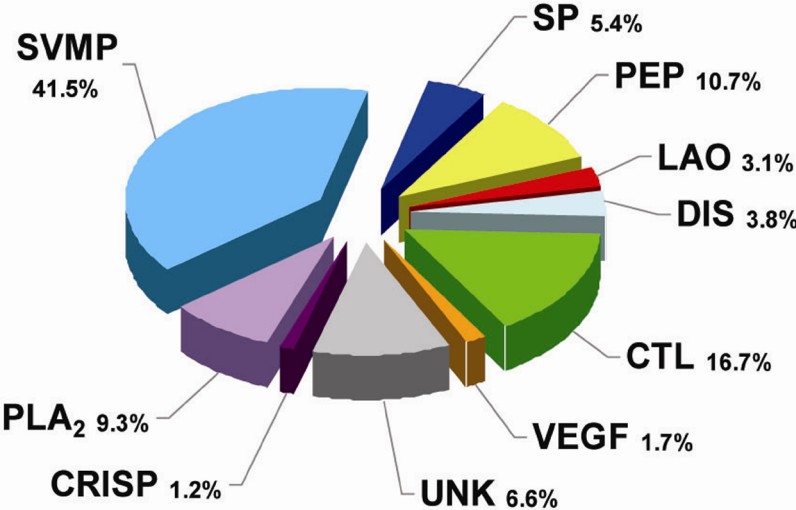

**Figure 2 Composition of *Bothrops punctatus* venom according to protein families, expressed as percentages of the total protein content.** SP, serine proteinase; $PLA_2$, phospholipase $A_2$; CRISP, cysteine-rich secretory protein; DIS, disintegrin; PEP, bradykinin-potentiating peptide-like (BPP-like); LAO, L-amino acid oxidases; SVMP, metalloproteinase; VEGF, vascular endothelium growth factor; CTL, C-type lectin/lectin-like; UNK, unknown/unidentified.

*ayerbei*, the venom of *B. punctatus* presents a high content of BPP-like peptides, strikingly differing from *B. asper* and *B. atrox* venoms in this regard. The possible trophic relevance of these vasoactive peptides among viperids remains elusive, and no clear correlations with prey types or habitats have been disclosed thus far. BPPs are oligopeptides of 5–14 amino acid residues, rich in proline residues and often presenting a pyroglutamate residue, which display bradykinin-potentiating activity. Their pharmacological effect is related to the inhibition of angiotensin I-converting enzyme (ACE) (*Ianzer et al., 2007*). Peak 4 of the HPLC separation of *B. punctatus* venom components (Fig. 1C) was identified as a BPP (Table 1), and its inhibitory activity on ACE was confirmed, showing a half-maximal inhibition of this enzyme at 0.9 mg/mL (Fig. 3A). Interest in snake venom BPPs stems from their potential in the development of hypotensive drugs, as exemplified by Captopril®. Overall, the comparison of *B. punctatus* venom with those of other pitvipers distributed in Colombia (Table 2) highlights the remarkable divergence of compositional profiles that have arisen through the evolution and diversification of snakes (*Casewell et al., 2013*).

The protein composition of *B. punctatus* venom correlates with the enzymatic activities assayed, as well as with those described in earlier studies (*Otero et al., 1992*; *Kuch et al., 1996*). L-amino acid oxidase (Fig. 3C), proteolytic (Fig. 4A), and $PLA_2$ (Fig. 4B) activities of this venom were corroborated. Interestingly, its proteolytic activity was higher than that of *B. asper* venom (Fig. 4A), and this might be related to the stronger hemorrhagic potency that was reported for *B. punctatus* venom in comparison to *B. asper* venom (*Otero et al., 1992*). Hemorrhage induced by viperid venoms is mainly dependent on the proteolytic action of SVMPs upon the microvasculature and its extracellular matrix support (*Bjarnason & Fox, 1994*; *Gutiérrez et al., 2005*), and this effect can be enhanced by venom components affecting haemostasis, such as procoagulant SPs with thrombin-like

**Table 2** **Comparison of the venom composition of *Bothrops punctatus* with venoms from pitviper species distributed in Colombia.**[*]

| Protein family | Snake species | | | | |
|---|---|---|---|---|---|
| | *Bothrops punctatus*[a] | *Bothrops atrox*[b] | *Bothrops asper*[c] | *Bothriechis schlegelii*[d] | *Bothrops ayerbei*[e] |
| Metalloproteinase | 41.5 | 48.5 | 44.0 | 17.7 | 53.7 |
| Phospholipase $A_2$ | 9.3 | 24.0 | 45.1 | 43.8 | 0.7 |
| Serine proteinase | 5.4 | 10.9 | 10.9 | 5.8 | 9.3 |
| BPP-like | 10.7 | 0.3 | - | 13.4 | 8.3 |
| CRISP | 1.2 | 2.6 | 0.1 | 2.1 | 1.1 |
| C-type lectin/lectin-like | 16.7 | 7.1 | 0.5 | - | 10.1 |
| VEGF | 1.7 | - | - | - | - |
| L-amino acid oxidase | 3.1 | 4.7 | 4.6 | 8.9 | 3.3 |
| Disintegrin | 3.8 | 1.7 | 1.4 | - | 2.3 |
| Kazal type inhibitor | - | - | - | 8.3 | - |
| Phosphodiesterase | - | - | - | - | 0.7 |
| Nerve growth factor | - | - | - | - | 0.1 |
| unknown | 6.6 | - | - | - | 1.7 |
| **Number of families** | **9** | **8** | **7** | **7** | |

**Notes.**

[*] Although *B. asper* and *B. schlegelii* are found in Colombia, data correspond to venoms from specimens found in Costa Rica.

[a] Present work.

[b] *Núñez et al. (2009)*.

[c] *Alape-Girón et al. (2008)*, specimens of Pacific versant.

[d] *Lomonte et al. (2008)*.

[e] *Mora-Obando et al. (2014)*.

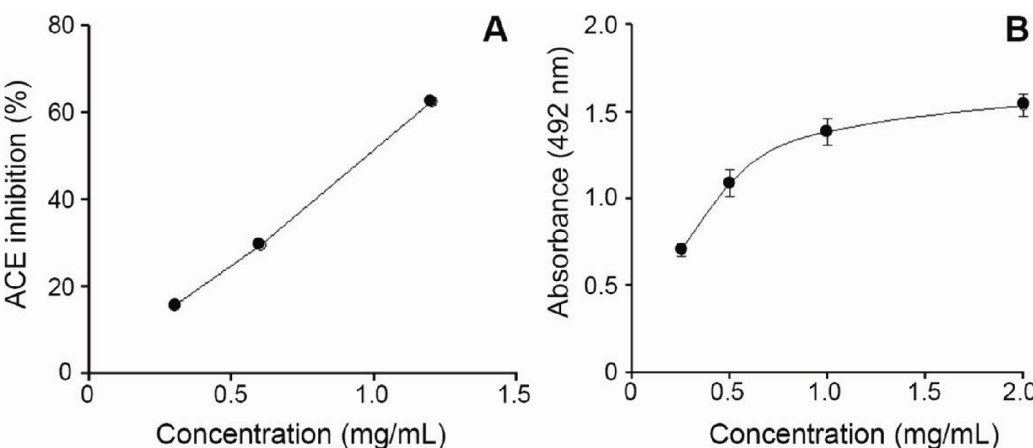

**Figure 3** **Bothrops punctatus venom activities.** (A) Inhibition of angiotensin-converting enzyme (ACE) by peak 4 of *B. punctatus* venom, identified as a BPP-like peptide (Table 1). Each point represents the mean ± SD of three replicates. (B) L-amino acid oxidase activity of *B. punctatus* venom. Each point represents the mean ± SD of three replicates.

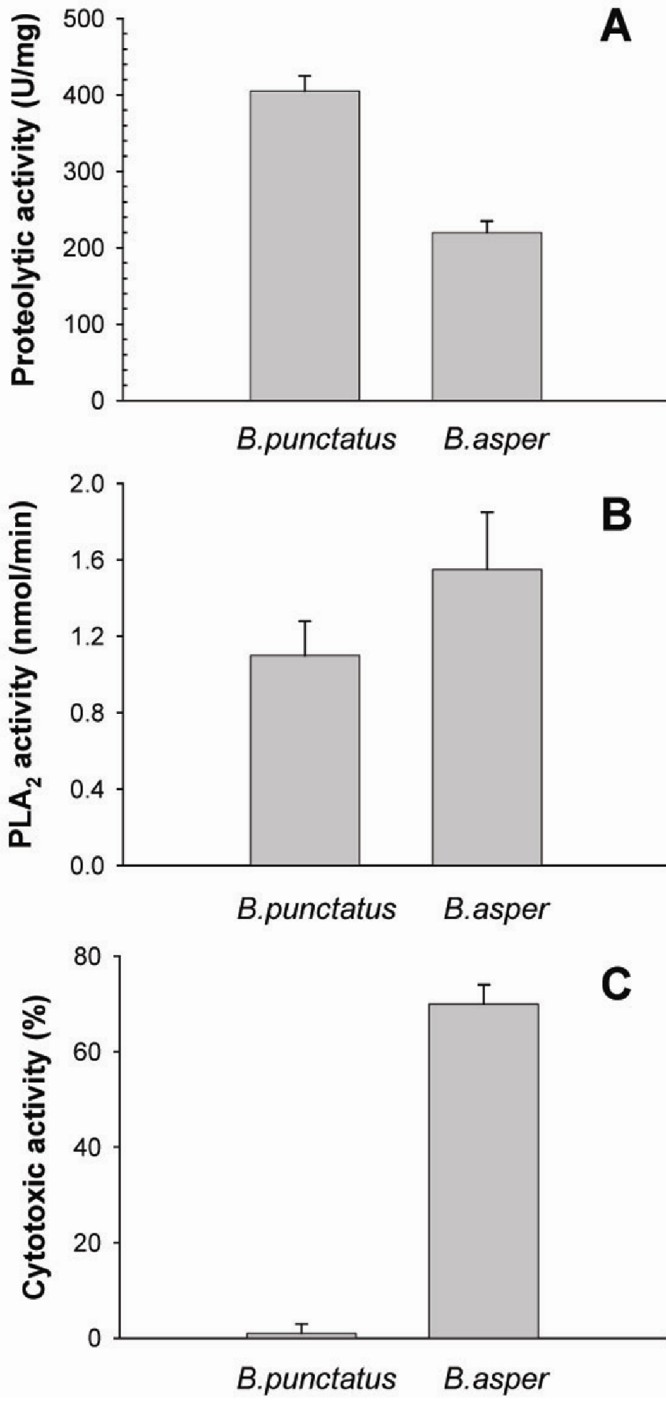

**Figure 4 Proteolytic (A), phospholipase A$_2$ (B), and cytotoxic (C) activities of *Bothrops punctatus* venom, compared to the venom of *Bothrops asper*.** Proteolytic activity was determined on azocasein, using 20 μg of each venom. Phospholipase A$_2$ activity was determined on 4-nitro-3-octanoyloxy-benzoic acid, using 20 μg of each venom. Cytotoxic activity was determined on C2C12 murine myoblasts, using 40 μg of each venom, as described in Methods. Bars represent mean ± SD of three replicates. For each activity, differences between the two venoms were significant ($p < 0.05$).

activity, or some CTL components and disintegrins that potently interfere with platelets, among others (*Gutiérrez, Escalante & Rucavado, 2009*; *Calvete et al., 2005*). Considering that the proportion of SVMPs is lower in *B. punctatus* than in *B. asper* venom (Table 2), the higher hemorrhagic action reported for the former (*Otero et al., 1992*) suggests that its abundant CTL components (16.7%) might include toxins that affect platelets, a hypothesis that deserves future investigation. On the other hand, the PLA$_2$ activity of *B. punctatus* venom was lower than that of *B. asper* (Fig. 4B), in agreement with their corresponding relative contents of these enzymes (Table 2). However, a major contrast was evidenced in the cytotoxic activity of these two venoms upon myogenic cells in culture, *B. punctatus* being essentially devoid of this effect, while *B. asper* causing overt cytolysis and LDH release under identical conditions (Fig. 4C). Since cytolysis of myogenic cells, an *in vitro* correlate for *in vivo* myotoxicity (*Lomonte et al., 1999*), has been shown to be mediated mainly by basic PLA$_2$s in the case of viperid venoms (*Gutiérrez & Lomonte, 1995*; *Lomonte & Rangel, 2012*), this finding anticipates that the catalytically active (D49) PLA$_2$s present in *B. punctatus* venom are likely to belong to the acidic type of these enzymes, which despite frequently having higher enzymatic activity than their basic counterparts, usually display very low, or even no toxicity (*Fernández et al., 2010*; *Van der Laat et al., 2013*). In contrast, the venom of *B. asper* is rich in basic D49 and K49 PLA$_2$s/PLA$_2$ homologues with strong cytolytic and myotoxic effects (*Angulo & Lomonte, 2005*; *Angulo & Lomonte, 2009*) that would explain the present findings. Although at least one PLA$_2$ component of *B. punctatus* venom was shown to belong to the K49 type of catalytically-inactive, basic PLA$_2$ homologues (fraction 23–25a; Table 1), its low abundance (1.3%) in the venom would be in agreement with the observed lack of cytotoxicity (Fig. 4C).

In summary, the general compositional profile of *B. punctatus* venom was obtained through the analytical strategy known as 'snake venomics'. The present data add to the growing body of knowledge on the remarkable diversity of compositional strategies in snake venom 'cocktails', in spite of the reduced number of gene families that encode their proteins/toxins (*Casewell et al., 2013*; *Calvete, 2013*). Due to the key adaptive role of venoms, this knowledge, in combination with toxicological, ecological, and natural history information, could lead to a deeper understanding of the evolutionary trends and selective advantages conferred by particular venom compositions in the divergence of snakes. In addition, compositional data may offer a more comprehensive basis to foresee the features of envenomings by this pitviper species, largely unreported in the literature.

## ACKNOWLEDGEMENTS

We thank Leidy Gómez, Paola Rey, and Wan-Chih Tsai for their valuable collaboration in the laboratory.

### Funding

This work was funded by grants obtained from Universidad de Antioquia Sostenibilidad (2013–2014), and from Vicerrectoría de Investigación, Universidad de Costa Rica (project

"Network for proteomic characterization of snake venoms of medical and biological relevance in Latin America"; 741-B3-760). The Proteomics Laboratory of the Instituto Clodomiro Picado is partially supported by the Vicerrectoría de Investigación, UCR. Maritza Fernández received a Young Researcher Fellowship from COLCIENCIAS. The funders had no role in study design, data collection and analysis, decision to publish, or preparation of the manuscript.

### Grant Disclosures
The following grant information was disclosed by the authors:
Universidad de Antioquia Sostenibilidad (2013–2014).
Vicerrectoría de Investigación, Universidad de Costa Rica.

### Competing Interests
Bruno Lomonte is an Academic Editor for PeerJ. There are no other competing interests to declare regarding this manuscript.

### Author Contributions
- Maritza Fernández Culma and Jaime Andrés Pereañez performed the experiments, analyzed the data, wrote the paper.
- Vitelbina Núñez Rangel conceived and designed the experiments, analyzed the data, wrote the paper.
- Bruno Lomonte analyzed the data, wrote the paper.

### Animal Ethics
The following information was supplied relating to ethical approvals (i.e., approving body and any reference numbers):

The "Programa de Ofidismo/Escorpionismo" has institutional permission from Universidad de Antioquia to have a serpentarium to keep live snakes for scientific research purposes.

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
