# Peer review of "Snake venomics of Bothrops punctatus, a semiarboreal pitviper species from Antioquia, Colombia"

_PeerJ, doi:10.7717/peerj.246_

## Round 0.1 · original submission · Minor Revisions

Please address critical points raised by the reviewer #2.

Reviewer 1 ·

Basic reporting

The manuscript meets the guidelines for this publisher.

Experimental design

Straightforward approach using typical proteomic techniques.

Validity of the findings

The results support the conclusions of this work.

Additional comments

The authors have described a straightforward analysis of the venom proteome of an interesting Bothrops snake. The work was well done and thorough with appropriate reference to the literature. Although little new scientific information results from this it does add to the general data base of venom proteomics and will potentially play a significant role as we continue studying these venoms.

·

Basic reporting

No Comments

Experimental design

No Comments

Validity of the findings

No Comments

Additional comments

Fernández-Culma and colleagues report a comprehensive proteomics characterization of the poorly characterized venom of Bothrops punctatus, an endangered, semi-arboreal pitviper species distributed in Panamá, Colombia, and Ecuador. The protein composition of B. punctatus venom comprises proteins from at least nine snake toxin families, and correlates with the enzymatic activities assayed. A comparison of B. punctatus venom and venoms from other Bothrops species evidenced notable differences. Overall, this well-designed and clearly written venomics study adds further data in support of the wide diversity of strategies that have evolved in snake venoms to subdue prey. In previous studies it was found that the lethal potency of this venom to mice was highest among the different Bothrops venoms analyzed, being only second to that of Crotalus durissus terrificus venom. Although reports of proven envenomings caused by B. punctatus are rare, a knowledge of its venom composition and toxic activities is of outmost relevance in case of having to attend an envenoming caused by this species. The manuscript deserves thus publication. However, the authors may wish to address the following minor points in a revised version:

1.- Lines 197-198: "BPPs are oligopeptides of 5–14 amino acid residues, rich in pyroglutamyl and proline residues..." Well, pyroglutamate occurs only once in a peptide sequence, and thus can not be considered an abundant residue....
2.- Lines 215-217: "...venom components affecting haemostasis, such as procoagulant SPs with thrombin-like activity, or some CTL components that potently interfere with platelets, among others." What about disintegrins?
3.- High-resolution Figure 2 should be provided.
4.- In the heading of Table 1, the symbol for "non-reduced" is missing.
5.- Whats the meaning of "pa" in EpaNGDVVCGGDDPCBK?
6.- Table 1, peaks 3 and 4. In many venoms these peaks contain tripeptide inhibitors of SVMPs, such as ZBW and ZNW. Did the authors check the mass range m/z 400-450?
7.- Why was 17b DNBDTYDXBYWFYGAK assigned as D49-PLA2? This peptide does not encompass the amino acid containing the D49 residue..

---

## Round 0.2 · accepted · Accept

Thank you for addressing reviewers' critical points and careful revision of the manuscript.